# Redundant roles of EGFR ligands in the ERK activation waves during collective cell migration

Shuhao Lin[1], Daiki Hirayama[2], Gembu Maryu[3] , Kimiya Matsuda[2], Naoya Hino[2], Eriko Deguchi[1], Kazuhiro Aoki[3,4,5] , Ryo Iwamoto[6], Kenta Terai[1] , Michiyuki Matsuda[1,2,7]

**Epidermal growth factor receptor (EGFR) plays a pivotal role in collective cell migration by mediating cell-to-cell propagation of extracellular signal-regulated kinase (ERK) activation. Here, we aimed to determine which EGFR ligands mediate the ERK activation waves. We found that epidermal growth factor (*EGF*)–deficient cells exhibited lower basal ERK activity than the cells deficient in heparin-binding EGF (*HBEGF*), transforming growth factor alpha (*TGFα*) or epiregulin (*EREG*), but all cell lines deficient in a single EGFR ligand retained the ERK activation waves. Surprisingly, ERK activation waves were markedly suppressed, albeit incompletely, only when all four EGFR ligands were knocked out. Re-expression of the EGFR ligands revealed that all but HBEGF could restore the ERK activation waves. Aiming at complete elimination of the ERK activation waves, we further attempted to knockout *NRG1*, a ligand for ErbB3 and ErbB4, and found that *NRG1*-deficiency induced growth arrest in the absence of all four EGFR ligand genes. Collectively, these results showed that EGFR ligands exhibit remarkable redundancy in the propagation of ERK activation waves during collective cell migration.**

## Introduction

Collective cell migration in mammalian tissues is a well-orchestrated cell movement underlying fundamental biological processes (Friedl & Gilmour, 2009; Mayor & Etienne-Manneville, 2016). MDCK cells, which hold clear apical-basolateral polarity and clear cell junctions (Dukes et al, 2011), are frequently used as a model of collective cell migration (Reffay et al, 2014; Das et al, 2015). The epidermal growth factor receptor (EGFR)-extracellular signal-regulated kinase (ERK) signaling cascade plays a pivotal role in the collective cell migration of various cell types including MDCK cells (Yarden & Sliwkowski, 2001; Friedl & Gilmour, 2009). During collective cell migration of MDCK cells, ERK activation propagates as multiple waves from the leader cells to

the follower cells in an EGFR-dependent manner (Hiratsuka et al, 2015; Aoki et al, 2017; Hino et al, 2020; Boocock et al, 2021).

EGFR is bound to and activated by a family of ligands that include epidermal growth factor (EGF), transforming growth factor alpha (TGFα), heparin-binding EGF-like growth factor (HBEGF), amphiregulin, betacellulin, epiregulin (EREG), and epigen (Harris et al, 2003). Extensive research has clarified the difference among the EGFR ligands with respect to binding affinity to four ErbB-family receptors including EGFR/ErbB1, sensitivity to proteases, subcellular localization, bioactivity to promote cell growth, and migration (Wilson et al, 2009; Singh et al, 2016). However, there are still many questions to be answered about the roles played by the endogenous EGFR ligands, because much of the current knowledge is based on exogenous bolus application of EGFR ligands to tissue culture cells. Meanwhile, knockout mice deficient in each of the seven EGFR ligand genes are viable and fertile (Schneider et al, 2008; Nanba et al, 2013), suggesting functional redundancy among the EGFR ligands (Riese & Cullum, 2014; Singh & Coffey, 2014; Taylor et al, 2014; Zeng & Harris, 2014). Therefore, abrogation of multiple EGFR ligands is essential to clearly demonstrate the activity of the endogenous EGFR ligands.

Here, to determine which EGFR ligand mediates the propagation of the ERK activation waves during collective cell migration of MDCK cells, we knocked out all four EGFR ligands expressed in MDCK cells, EGF, TGFα, HBEGF, and EREG. We found that propagation of the ERK activation waves was markedly suppressed only when all four EGFR ligands were knocked out. Re-expression of each EGFR ligand showed that EGF, TGFα, and EREG, but not HBEGF, can restore the ERK activation waves.

## Results

### A new FRET biosensor for ERK without cross-reactivity to CDK1

In previous studies (Aoki et al, 2013, 2017; Hino et al, 2020), we used the FRET-based EKAREV biosensor for the detection of the ERK activation waves. However, EKAREV responds not only to ERK, but

[1]Department of Pathology and Biology of Diseases, Graduate School of Medicine, Kyoto University, Kyoto, Japan   [2]Research Center for Dynamic Living Systems, Graduate School of Biostudies, Kyoto University, Kyoto, Japan   [3]Division of Quantitative Biology, National Institute for Basic Biology, National Institutes of Natural Sciences, Okazaki, Japan   [4]Quantitative Biology Research Group, Exploratory Research Center on Life and Living Systems (ExCELLS), National Institutes of Natural Sciences, Okazaki, Japan   [5]Department of Basic Biology, School of Life Science, SOKENDAI (The Graduate University for Advanced Studies), Okazaki, Japan   [6]Research Institute for Microbial Diseases, Osaka University, Osaka, Japan   [7]Institute for Integrated Cell-Material Sciences, Kyoto University, Kyoto, Japan

Correspondence: matsuda.michiyuki.2c@kyoto-u.ac.jp

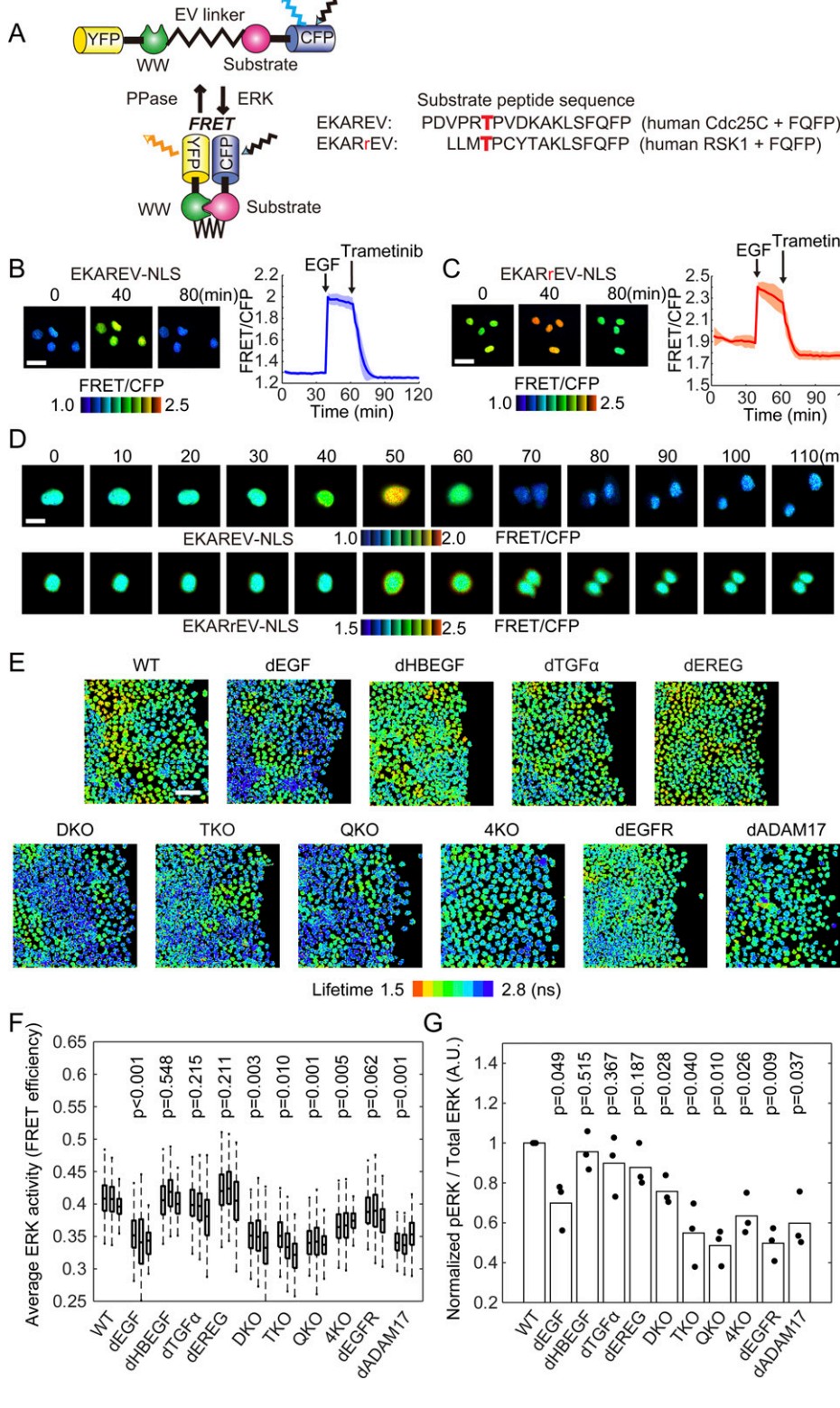

**Figure 1. Properties of EKARrEV-NLS biosensor and ERK activity of MDCK cell lines after knockout of epidermal growth factor receptor ligand genes.**

**(A)** Mode of action of the intramolecular FRET biosensors for ERK, EKAREV, and EKARrEV. The red character T indicates the phosphorylation site. **(B, C)** MDCK cells expressing EKAREV-NLS (B) or EKARrEV-NLS (C) were observed under a wide-field fluorescence microscope to acquired FRET/CFP time-lapse images. During imaging, 10 ng mL$^{-1}$ epidermal growth factor and 200 nM trametinib were added. Representative ratio images are shown in the IMD mode. Line plots show time courses of the FRET/CFP ratio for four cells from a single experiment. Solid lines represent the means; shaded areas represent SD. Scale bar, 40 $\mu$m. **(D)** MDCK cells expressing EKAREV-NLS (top) or EKARrEV-NLS (bottom) were imaged during mitosis, shown in the IMD mode. Scale bar, 20 $\mu$m. **(E)** Lifetime images of the donor fluorescence was acquired by a confocal microscope equipped with a 440 nm ps pulsed diode laser. MDCK cells are WT, single gene knockout (dEGF, dHBEGF, dTGFα, and dEREG), *EGF*/*HBEGF* double knockout (DKO), *EGF*/*HBEGF*/*TGFα* triple knockout (TKO), *EGF*/*HBEGF*/*TGFα*/*EREG* quadruple knockout (QKO and 4KO), *EGFR* knockout (dEGFR), and *ADAM17* knockout (dADAM17). **(F)** Cells are time-lapse imaged every 10 min for 100 min. FRET efficiency of randomly chosen cells was plotted for each cell line. The total number of analyzed cells from two independent experiments is as follows: WT, 271, 231, and 201 cells; dEGF, 250, 255, and 171 cells; dHBEGF, 260, 136, and 206 cells; dTGFα, 227, 163, and 204 cells; dEREG, 236, 155, and 213 cells; DKO, 311, 241, and 293 cells; TKO, 253, 229, and 189 cells; QKO, 263, 232, and 186 cells; 4KO, 252, 255, and 179 cells; dEGFR, 217, 155, and 153 cells; dADAM17, 146, 140, and 198 cells. **(G)** 9 h after the removal of silicone confinement, MDCK cells were analyzed by immunoblotting with anti–phospho-ERK or anti–pan-ERK antibody. The phospho-ERK signal normalized to the pan-ERK signal is shown. Data from three independent experiments are shown. **(F, G)** *P*-values of two-tailed *t* test are shown on the top of panels (F) and (G).

also to cyclin-dependent kinase 1 (CDK1), showing increases in FRET/CFP signal during mitosis even in the presence of an MEK inhibitor. To overcome this flaw, we developed a new ERK biosensor named EKARrEV by replacing the substrate peptide derived from Cdc25C with that from ribosomal protein S6 kinase A1 (RSK1) (Fig 1A). In MDCK cells expressing either EKAREV-NLS (with a nuclear localization signal) or EKARrEV-NLS, the FRET/CFP ratio was increased by EGF and decreased by the MEK inhibitor

trametinib (Fig 1B and C). In EKAREV-NLS–expressing cells, but not in EKARrEV-NLS–expressing cells, the FRET/CFP ratio increased immediately before mitosis (Fig 1D). Although EKARrEV-NLS exhibited a smaller dynamic range and higher basal FRET/CFP ratio than the prototype EKAREV-NLS, we adopted EKARrEV-NLS because the increase in the FRET/CFP ratio during mitosis hampers automatic image analysis of ERK activation waves. It should be noted that Ponsioen et al (2021) also recently reported another EKAREV-derived biosensor by modifying the substrate peptide to eliminate the cross-reactivity to CDK1 (Ponsioen et al, 2021).

### Dependency on EGF for the mean ERK activity in serum-starved MDCK cells

Among the seven known EGFR ligands, EGF, HBEGF, TGFα, and EREG were detected by RNA-Seq analysis of MDCK cells (Fig S1A) (Shukla et al, 2015). The expression of amphiregulin is marginal; therefore, we did not further pursue its role. To examine the roles of these EGFR ligands in the propagation of ERK activation waves, we employed CRISPR/Cas9-mediated knockout of each EGFR ligand gene (Fig S2A). The resulting cell lines with knockout of a single EGFR ligand were named MDCK-dEGF, MDCK-dHBEGF, MDCK-dTGFα, and MDCK-dEREG, or simply dEGF, dHBEGF, dTGFα, and dEREG hereinafter. Because none of them exhibited a detectable decrease in the ERK activation waves, we knocked out the EGFR ligands sequentially from two abundantly expressed genes, *EGF* and *HBEGF*. Because the ERK activation waves were clearly visible in the double knockout cells, which we named DKO, *TGFα* was further knocked out to obtain triple knockout cells, TKO, and then *EREG* was knocked out to obtain quadruple knockout cells, QKO. After finding marked suppression of the ERK activation waves in QKO, all four genes were knocked out simultaneously by introducing four gRNAs. The new clone deficient from the four EGFR ligand genes was designated as 4KO. We did not find significant change in the mRNA expression levels of the other three EGFR ligands and four neuregulins in either 4KO or QKO (Fig S1B). *EGFR* and *ADAM17* were also knocked out for comparison, generating the dEGFR and dADAM17 cell lines, respectively (Fig S2B). After single-cell cloning, frame-shift mutations of both alleles were confirmed by genome sequencing (Fig S2C). These cell lines were subjected to a confinement release assay to observed collective cell migration.

We first examined the ERK activity using population-based methods. The mean FRET efficiency of the biosensor was determined for each cell line by fluorescence lifetime microscopy (FLIM) (Fig 1E and F). Cells were also subjected to immunoblotting with anti-phospho-ERK antibody (Figs 1G and S1C). In both experiments, cell lines deficient from *EGF*, that is, dEGF, DKO, TKO, QKO, and 4KO, exhibited lower ERK activity than the wild type (WT) cells. This essential role of EGF in the mean ERK activity was confirmed by establishing another clone that is deficient in *EGF* (Fig S1D). The level of ERK activity in these *EGF*-deficient cell lines was comparable to that in dEGFR cells, indicating that EGF is the principal endogenous EGFR ligand maintaining the basal EGFR activity in serum-starved MDCK cells. Of note, EKARrEV reflects the balance between ERK activity and phosphatase activity, whereas the anti-phospho-ERK antibody detects the phosphorylation of the catalytic loop of ERK. The discrepancy in the ERK activity of dEGFR cells between fluorescence lifetime imaging

microscopy and immunoblotting may arise from this difference. ADAM17 deficiency also decreased the average ERK activity, indicating that shedding of growth factors is required for the ERK activation.

### Redundant roles of EGFR ligands in the propagation of ERK activation waves during collective cell migration as revealed by single cell analysis

For the analysis of collective cell migration, MDCK cells expressing EKARrEV-NLS were seeded 1 d before imaging within a culture-insert. After the removal of the culture-insert, cells were imaged in serum-free medium for more than 12 h (Fig 2A and Video 1). In this study, most experiments, unless noted otherwise, were performed in the absence of serum to exclude the effect of serum-derived growth factors. To evaluate the migration speed, cells that were located less than 100 µm from the leading edge at the time of confinement release were classified as the leader and submarginal cells (Fig 2B). We analyzed these cells together because the leader cells are often replaced by the following submarginal cells during long-term imaging. After single-cell tracking analysis, the displacement of the leader and submarginal cells before and 12 h after the start of collective cell migration was defined as the cell migration distance. The impairment of migration was clear when the two most abundant EGFR ligands, EGF and HBEGF, were absent, suggesting that the total amount of EGFR ligands may be the primary determinant for the migration of the leader and submarginal cells.

We previously reported that ERK activation waves from the leader cells are indispensable for the directed migration of the follower cells (Aoki et al, 2017; Hino et al, 2020). Therefore, the effect of EGFR ligand deficiency on the propagation of the ERK activation waves was examined hereafter. In WT cells, clear ERK activation waves were propagated from the leader cells to the follower cells (Video 1). Similarly, clear ERK activation waves were observed in single knockout cells, DKO, and even TKO cells. Only when all four EGFR ligand genes were knocked out in QKO and 4KO were the ERK activation waves markedly suppressed. To quantitatively understand the contribution of each EGFR ligand to the propagation of the ERK activation waves, we employed single cell–based analysis of the follower cells (Fig 2C). After sine curve fitting of the time course of the FRET/CFP ratio in each cell (Hiratsuka et al, 2015), three parameters—basal ERK activity, amplitude, and duration—were obtained to evaluate the effect of EGFR ligand deficiency. Basal ERK activity was decreased in the *EGF*-deficient cell lines, dEGF, DKO, TKO, QKO, and 4KO (Fig 2D), as observed in the population-based analysis (Fig 1E–G). The basal ERK activity was decreased in dADAM17, but not dEGFR. The decrease of amplitude was obvious when all four EGFR ligand genes were knocked out (Fig 2E). Negligible changes of duration period were observed in all cell lines, although the variance was markedly larger in QKO, 4KO, and dEGFR, probably because of the low amplitude of each pulse (Fig 2F). These results indicated that EGF is the primary EGFR ligand to maintain the basal ERK activity, whereas all EGFR ligands may contribute to the propagation of ERK activation.

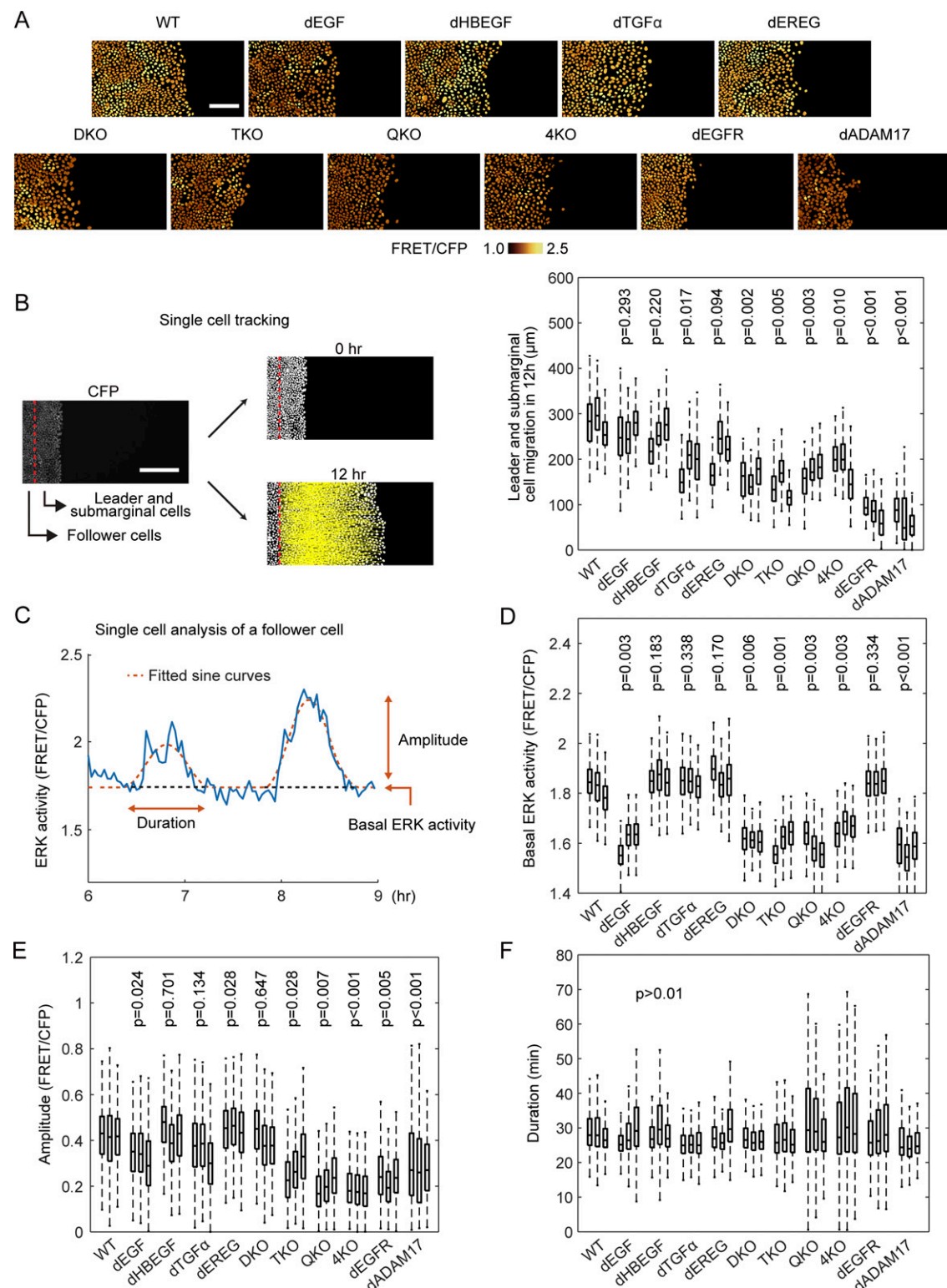

**Figure 2. Single-cell analyses of ERK activation dynamics in MDCK cells deficient of epidermal growth factor receptor ligands.**
MDCK cells expressing EKARrEV were subjected to confinement release assay. **(A)** Time-lapse images of FRET/CFP ratio were acquired every 2 min for up to 12 h to generate Video 1. Shown are snapshots cropped from the Video. Scale bar, 100 μm. **(B)** We defined the cells locating within 100 μm from the edge of the open space as the leader and submarginal cells, the others as follower cells. CFP channel images were binarized for single cell detection. From the track of the leader and submarginal cells, migration distance in 12 h was measured (left). Scale bar, 200 μm. Boxplot of leader and submarginal cell migration in 12 h observation (right). The total number of analyzed cells from three independent experiments is as follows: WT, 395, 167, and 298 cells; dEGF, 386, 408, and 178 cells; dHBEGF, 447, 370, and 478 cells; dTGFα, 408, 366, and 388 cells; dEREG, 505, 525, and 504 cells; DKO, 256, 233, and 304 cells; TKO, 197, 270, and 229 cells; QKO, 278, 463, and 441 cells; 4KO, 407, 233, and 154 cells; dEGFR, 584, 303, and 217 cells; dADAM17, 254, 183, and 205 cells. **(C)** After applying a mask for nuclei of the follower

### Redundant roles of EGFR ligands in the propagation of ERK activation waves as revealed by heat map analysis

We then employed population-based analysis of ERK activation waves. First, a heat map of the ERK activity was obtained by interpolation of the FRET/CFP images (Fig 3A). The directedness and the number of waves were analyzed by particle image velocimetry (PIV) and kymograph, respectively. We sometimes observed waves propagating in random directions, particularly at the area remote from the leader cells. PIV analysis showed that the directionality of ERK activation waves from the leader cells to the follower cells was markedly perturbed in QKO, 4KO, and dEGFR cells (Fig 3B). Similarly, the number of ERK activation waves was markedly decreased in QKO, 4KO, dEGFR, and dADAM17 cells (Fig 3C). Among the cell lines, we did not observe apparent changes in the speed of the ERK activation waves when they appeared.

We previously showed that the ERK activation waves are induced by mechanical stretch from the leader cells, which retain high ERK activity (Hino et al, 2020). Therefore, the lack of ERK activation waves in QKO and 4KO cells may be due to the low ERK activity in the leader cells. To eliminate this possibility, we employed a rapamycin-activable (RA) system combined with mSos1, a Ras guanine nucleotide exchange factor (Aoki et al, 2017). MDCK cells carrying RA-mSos1 were seeded next to MDCK WT or 4KO cells (Fig 3D). Upon rapamycin treatment, ERK was activated in the RA-SOS cells, followed by the emergence of ERK activation waves in WT cells, but not in 4KO cells, at the interface (Fig 3E and F and Video 2), indicating that ERK activation in RA-mSOS1 cells cannot be transmitted to the neighboring 4KO cells. In short, none of the knockouts of single EGFR ligands affected either the collective cell migration or the propagation of ERK activation waves, which were markedly impaired in cell lines deficient in all four EGFR ligand genes.

### Restoration of ERK activation waves by re-expression of the EGFR ligands

Do all EGFR ligands contribute to the ERK activation waves? To answer this question, we re-expressed each of the four EGFR ligands in 4KO cells and performed the confinement release assay (Fig 4A and Video 3). The results showed that all EGFR ligands accelerated the migration of the leader and submarginal cells (Fig 4B), but none of them increased the basal ERK activity to the level in the WT cells (Fig 4C). Interestingly, except HBEGF, all EGFR ligands restored the ERK activation waves, as evidenced by the increase in the number, amplitude, and directionality of waves from the leader cells (Fig 4D–F). We also quantified the area of ERK activation waves by using binarized interpolated FRET/CFP ratio images (Fig 4G). Again, except HBEGF, all EGFR ligands were able to restore the area of ERK activation waves.

Next, we examined whether bath application of EGFR ligands could also restore the ERK activation waves. EGFR ligands at the same concentration, 10 ng ml$^{-1}$ (~2 nM), were added to 4KO cell lines after the removal of the confinement (Fig S3A and Video 4). All EGFR ligands induced transient ERK activation and promoted leader cell migration

(Fig S3B and C). Nevertheless, none were able to induce ERK activation waves from the leader cells, indicating that only the endogenous EGFR ligands can generate ERK activation waves from the leader cells. Notably, the duration of the transient ERK activation was markedly longer in HBEGF-treated cells than in the cells treated with EGF, TGFα, or EREG (Fig S3D). To exclude the possibility that EGF ligands had been exhausted, EGF-stimulated cells were restimulated by EGFR ligands (Fig S3E). ERK was activated only modestly, suggesting that the decreased ERK activity is primarily caused by the down-regulation of EGFR.

### Growth arrest induced by the knockout of *NRG1* in the absence of all four EGFR ligands

Albeit much less intensely than in WT cells, the ERK activation waves from the leader cells persisted even in QKO, 4KO, and dEGFR cells (Fig 2A and Video 1), implying the involvement of neuregulins (NRGs), the ligands for ErbB3 and ErbB4. Among the four *NRG* genes, only the expression of NRG1 was detected in MDCK cells (Fig S1). Therefore, we attempted to knockout *NRG1* in 4KO cells with three different single guide RNAs (sgRNAs) targeted to exon 6 or exon 9 of *NRG1*. Among 75 individual clones, 33 clones exhibited heterozygous deletion/insertion, but none showed homozygous knockout, suggesting that additional knockout of *NRG1* in 4KO cells induces growth retardation. Therefore, we expressed NRG1 cDNA flanked by the *loxP* sequence before knockout of the endogenous *NRG1* gene (Fig 5A). This procedure successfully generated a clone named 5KO-loxP-NRG1, in which two-base deletion and single-base insertion were found in the *NRG1* alleles (Fig S2C). We further expressed CreERT2 to generate 5KO-loxP-NRG1-CreERT2 cells and compared the cell growth in the presence and absence of 4-hydroxy-tamoxifen (4-OHT). As expected, after the addition of 4-OHT, 5KO-loxP-NRG1-CreERT2 cells exhibited cell growth arrest (Fig 5B), demonstrating the essential role of NRG1 in the growth of 4KO cells and indispensable role of the autocrine activation of the ErbB-family receptors in the growth of MDCK cells. Finally, we addressed whether the EGFR ligands can be supplemented in trans. When co-cultured with the WT cells, 5KO-loxP-NRG1-CreERT2 cells grew in the presence of 4-OHT as rapidly as in the absence of 4-OHT (Fig 5C), indicating that EGFR ligands secreted from the WT cells can support the growth of *NRG1*-deficient 4KO cells. Interestingly, however, bolus addition of EGF into the medium failed to cancel the effect of *NRG1* deficiency caused by 4-OHT (Fig 5D). Thus, paracrine activation of EGFR receptors is indispensable for the survival of MDCK cells.

## Discussion

To examine the contribution of each EGFR ligand to the ERK activation waves, we first started to knockout the EGFR ligand genes from abundantly expressed *EGF* and *HBEGF*, followed by *TGFα* and *EREG*. Against our expectation, we found that the ERK activation waves from the leader cells were markedly inhibited only when all

---

cells, the time course of FRET/CFP ratio in each cell was analyzed from six to 9 h after the start of migration (blue line). The ratio values were fitted by sine curves for the detection of waves (red line). **(D, E, F)** For each wave, basal ERK activity (D), amplitude (E) and duration (F) were determined. The numbers of analyzed cells are as follows: WT, 814, 801, and 990 cells; dEGF, 789, 596, and 789 cells; dHBEGF, 846, 617, and 683 cells; dTGFα, 808, 762, and 838 cells; dEREG, 762, 1,030, and 953 cells; DKO, 638, 656, and 560 cells; TKO, 649, 778, and 567 cells; QKO, 507, 638, and 567 cells; 4KO, 681, 291, and 451 cells; dEGFR, 643, 493, and 684 cells; dADAM17, 346, 498, and 318 cells. **(B, D, E, F)** P-values of two-tailed t test (WT to others) are shown on the top of panels (B, D, E, F).

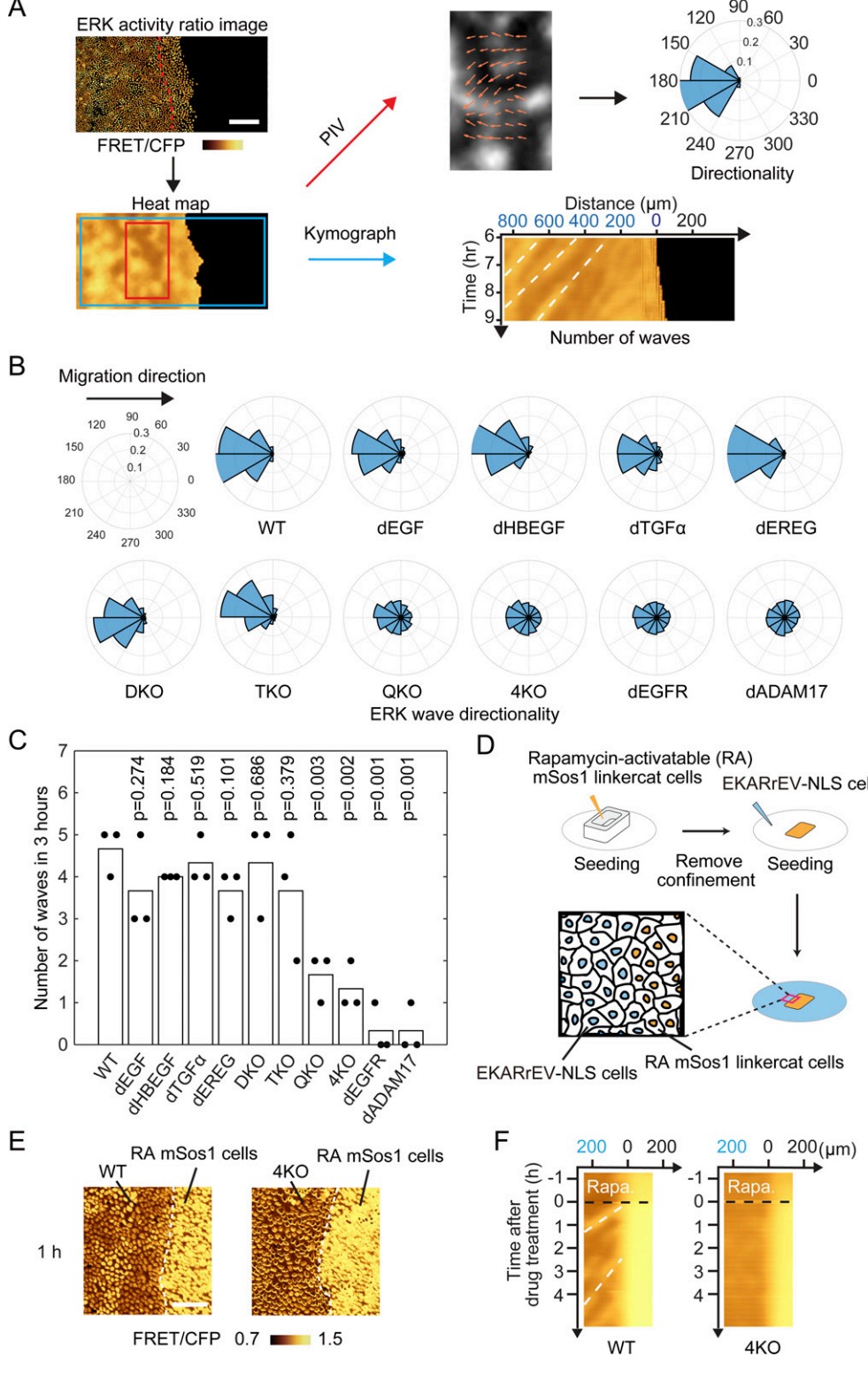

**Figure 3. Propagation of ERK activation waves from leader cells.**
**(A)** Heat maps of ERK activity were obtained by interpolating the signals between the nuclei of cells and used for particle image velocimetry and kymograph analysis. Directionality was measured by particle image velocimetry and shown with polar histogram. Kymograph was obtained by interpolated ratio image. White broken lines indicate representative ERK waves. Scale bar, 200 μm. **(B)** The direction of ERK activation wave from six to 9 h after releasing of silicone confinement was shown by Polar histograms. Shown are summation of three independent experiments. **(C)** During the period of 6–9 h after the release of silicone confinement, the number of ERK activation waves from the leader cells was manually counted on the kymograph. Each dot represents the number of counted ERK activation waves in an independent experiment. *P*-values of two-tailed *t* test (WT to the others) are shown on the top. **(D, E, F)** For the artificial generation of ERK activation waves, MDCK cells expressing EKARrEV and rapamycin-activable mSos1 was seeded in a silicone confinement. **(D)** MDCK cells expressing EKARrEV alone were seeded in the surrounding area (D). Upon rapamycin addition, ERK activation waves were generated at the border of cell lines. **(E)** Time-lapse FRET/CFP ratio images were acquired to generate Video 2. Shown are representative FRET/CFP ratio images 1 h after addition of rapamycin. White broken lines indicate the border of cell lines. Scale bar, 50 μm. **(F)** Kymographs of ERK activity in MDCK-WT and MDCK-4KO cells upon rapamycin addition. White broken lines indicate representative ERK waves.

four EGFR ligands were knocked out in QKO (Figs 2E and 3C). A potential problem of our gene knockout approach is that by this sequential knockout of the EGFR ligand genes, off-target mutations may accumulate. Therefore, we knocked out *HBEGF*, *EGF*, *TGFα*, and *EREG* simultaneously to obtain 4KO cells. Similarly, we used two independent clones of *EGF*-deficient cells. By using these independent clones, we minimized the potential effect of off-target mutation(s).

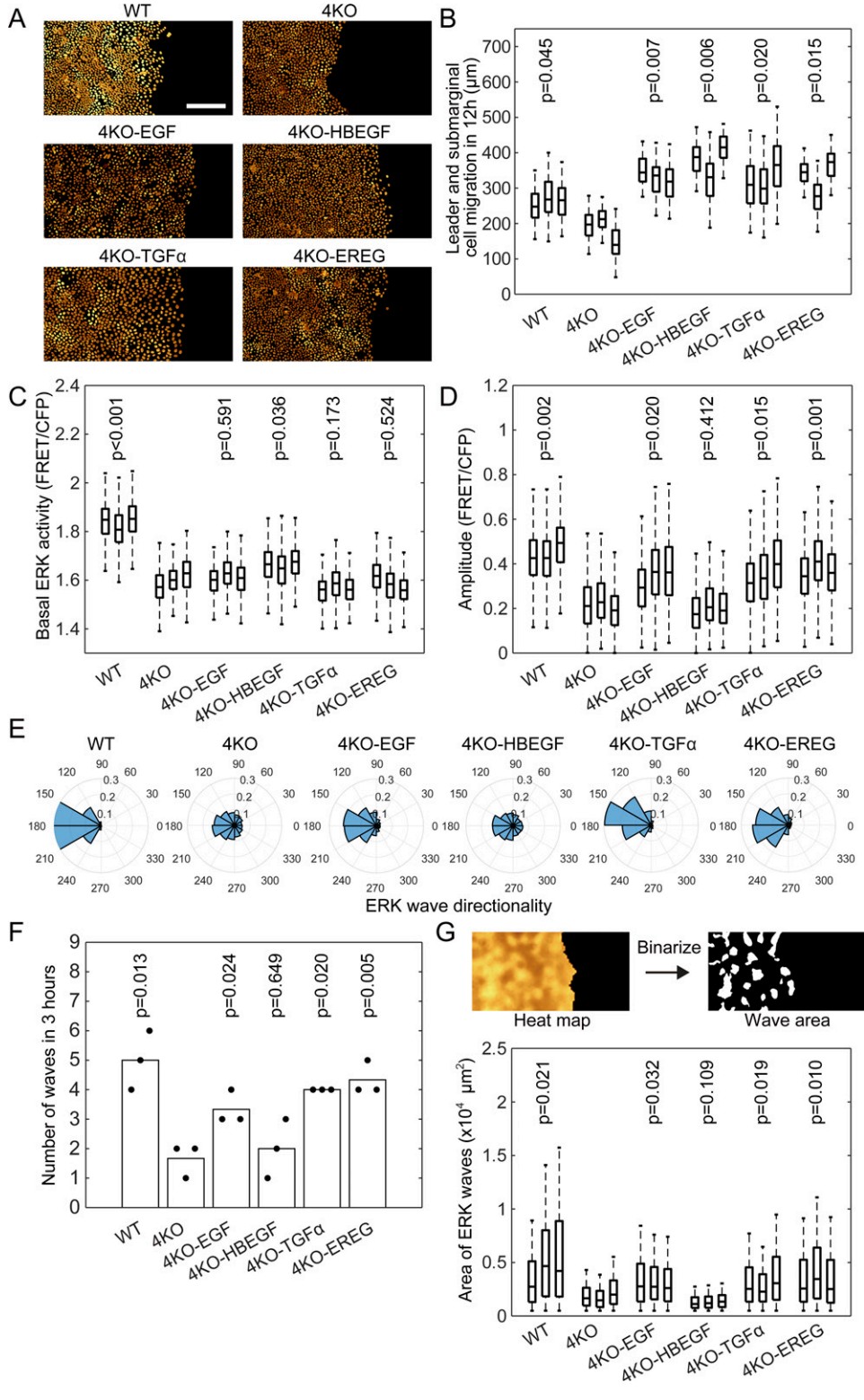

**Figure 4. Restoration of ERK activation waves by expression of epidermal growth factor receptor ligands in 4KO cells.**
The MDCK-4KO cells expressing the EKARrEV biosensor were transfected stably with an expression vector for cDNA of canine EGF, HBEGF, TGFα, or EREG; the resulting cell lines were named as 4KO-EGF, 4KO-HBEGF, 4KO-TGFα, and 4KO-EREG, respectively. WT, 4KO and epidermal growth factor receptor ligands expressing 4KO cells were subjected to confinement release assay. Three independent experiments were performed for each cell line. **(A)** Time-lapse FRET/CFP ratio images were acquired to generate Video 3. Shown are snapshots cropped from the Video. Scale bar, 200 μm. **(B)** Boxplot of leader and submarginal cell migration in 12 h observation. The number of analyzed cells from three independent experiments is as follows: WT, 357, 215, and 283 cells; 4KO, 348, 274, and 328 cells; 4KO-EGF, 130, 426, and 420 cells; 4KO-HBEGF, 224, 360, and 368 cells; 4KO-TGFα, 192, 300, and 215 cells; 4KO-EREG, 112, 404, and 373 cells. Data of three independent experiments are shown. **(C, D)** Box plots of basal ERK activity (C) and amplitude (D) of ERK activation in each cell line from six to 9 h after releasing of silicone confinement. The numbers of analyzed cells are as follows: WT, 703, 945, and 653 cells; 4KO, 656, 446, and 556 cells; 4KO-EGF, 576, 697, and 732 cells; 4KO-HBEGF, 667, 596, and 556 cells; 4KO-TGFα, 452, 638, and 484 cells; 4KO-EREG, 757, 786, and 902 cells. Data of three independent experiments are shown. **(E)** Polar histograms showing the distribution of ERK wave direction from 6–9 h after releasing of silicone confinement. Shown are summation of three independent experiments. **(F)** During the period of 6–9 h after the release of silicone confinement, the number of ERK activation waves from the leader cells was counted. Each dot represents the number of counted ERK activation waves in an independent experiment. **(G)** Wave area was measured after binarizing the denoised interpolated ratio image (upper). Box plots of the ERK wave area in each cell line at each frame from 6–9 h after releasing of silicone confinement. Data of three independent experiments are shown. **(B, C, D, F, G)** P-values of two-tailed t test (4KO to others) are shown in panels (B, C, D, F, and G).

Currently, we have not succeeded in eliminating the ERK activation waves by the knockout of EGFR ligand genes. This failure is because we could not examine the effect of NRG1 in 4KO cells due to the growth arrest (Fig 5B). To further pursue this issue, we probably need to establish MDCK cell variants that grow without input from receptor-type tyrosine kinases. Augmentation of cell survival signals such as the PI-3 kinase pathway may help to establish such cell lines.

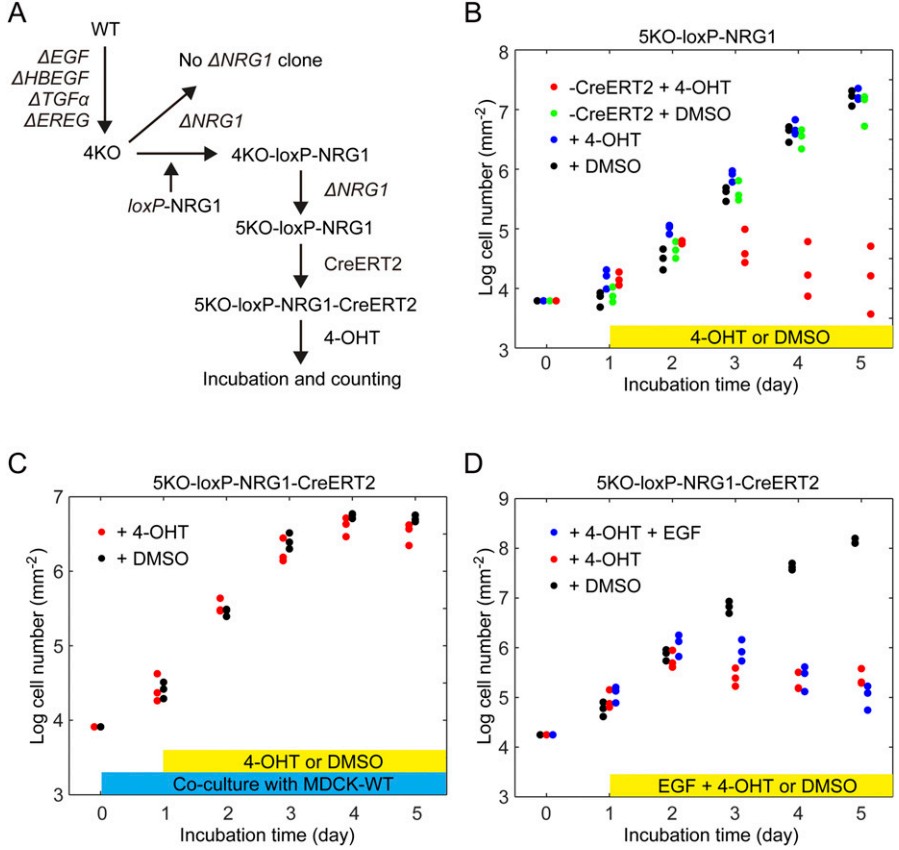

**Figure 5. Growth arrest of cells deficient from all epidermal growth factor receptor ligands.**
**(A)** Schematics of experiment to obtain 5KO cell line. The MDCK-4KO cells expressing EKARrEV biosensor were transfected stably with an expression vector for *loxP*-NRG1-*loxP* named as 4KO-loxP-NRG1. Then 5KO-loxP-NRG1 was generated by knocking out endogenous *NRG1*. Followed by integrating CreERT2, 5KO-loxP-NRG1-CreERT2 was treated with 4-OHT to obtain 5KO cell line. **(B)** The growth rate represented on a log (cell number) basis. Cell numbers was measured over 5 d post-seeding of 5KO-loxP-NRG1 cell line with addition of 4-OHT (blue dots) or DMSO (black dots) and 5KO-loxP-NRG1-CreERT2 cell line with addition of 4-OHT (red dots) or DMSO (green dots) at day one. Each dot represents cell number in an independent experiment. **(B, C)** 5KO-loxP-NRG1-CreERT2 cells were co-cultured with equal number of MDCK-WT cells and analyzed as in (B). **(B, D)** 5KO-loxP-NRG1-CreERT2 cells was analyzed as in (B) except that epidermal growth factor was exogenously added with 4-OHT.

We previously reported that a disintegrin and metalloprotease 17 (ADAM17), also known as tumor necrosis factor alpha converting enzyme (TACE), plays a critical role in the propagation of ERK activation waves in MDCK cells (Aoki et al, 2017; Hino et al, 2020). All pro-EGFR ligands except EGF are cleaved by ADAM17 to be soluble EGFR ligands (Sunnarborg et al, 2002). Then, why does EGF also restore the ERK activation waves? The ERK activation waves persist even in 4KO and QKO cells, suggesting the involvement of ErbB3 and ErbB4. EGF is the principal EGFR ligand that maintains basal ERK activity (Fig 2), which is probably required for efficient ADAM17 activation (Fan & Derynck, 1999). Therefore, by restoring EGF-mediated ERK activation, NRG1, a substrate of ADAM17, may efficiently contribute to the ERK activation propagation in an ErbB3 and ErbB4-dependent manner. Another related unanswered question is that of why HBEGF failed to restore the ERK activation waves. Because the migration of the leader and submarginal cells was accelerated by the HBEGF expression to the level of other EGFR ligands (Fig 4B), functional HBEGF was expressed in 4KO-HBEGF cells. Because the auto-regulatory role of the heparin-binding domain of HBEGF may prevent proteolytic release in MDCK cells (Takazaki et al, 2004; Prince et al, 2010), ADAM17 activation caused by ERK activation waves may not be sufficient for the cleavage of HBEGF in MDCK cells. Alternatively, the prolonged ERK activation by HBEGF, for about 2 h (Fig S3D), may render HBEGF inappropriate for the propagation of the ERK activation waves, because duration of ERK activation is approximately a half hour both in MDCK cells (Fig

2F) and in the mouse epidermis (Hiratsuka et al, 2015). Recently, in MCF10A human mammary epithelial cells, amphiregulin, expressed only at a marginal level in MDCK cells, was shown to mediate ERK activation from oncogene-expressing cells to neighboring normal cells (Aikin et al, 2020). Therefore, the dependency to each EGFR ligand and the redundancy appears cell type-specific.

Conventional immunoblotting analysis with anti-pERK antibodies measures the average ERK activity that reflects the basal ERK activity and the duration and amplitude of the ERK activation waves (Fig 2C). Only dEGF and dADAM17 exhibited low basal ERK activity (Fig 2D). This observation is counter-intuitive by the following three reasons. First, EGF is the only EGFR ligands in MDCK cells that is not cleaved by ADAM17. Second, cells deficient of *EGFR*, the sole EGF receptor, retains high basal ERK activity. Last but not the least, re-expression of *EGF* failed to restore the basal ERK activity (Fig 4C). The low basal ERK activity by *EGF* knockout was confirmed by establishing another *EGF*-knockout cell lines by deleting most of the exon 1 of *EGF* gene (Fig S1D). This enigma highlights the complexity of the regulation of ERK activity by EGFR ligands and awaits future study.

The 4KO and QKO cells will provide a versatile platform to highlight the differences among EGFR ligands in a physiological context. Previously, extensive characterization of EGFR ligands has been conducted by bath application to tissue culture cells. There are at least two serious problems in this approach. First, bath application of EGFR ligands may not evoke the physiological phenotypes by autocrine

stimulation. In fact, autocrine stimulation by EGFR ligands has been reported to promote greater cell migration of mammary epithelial cells compared to bath application (Joslin et al, 2007). In the present study, we also found that the ERK activation waves were restored by re-expression of EGFR ligands (Fig 4A), but not by the bath application of these EGFR ligands (Fig S3A). Second, the specific activity of each of the EGFR ligands used in previous studies might be significantly different. In one study using normal human epidermal keratinocytes, 2 nM EREG induced ERK phosphorylation more strongly than 10 nM EGF (Draper et al, 2003), whereas in another study using MCF-7 mammary cancer cells, 10 $\mu$M EREG and 16 nM EGF induced ERK phosphorylation to a similar level (Freed et al, 2017). In the present study, 10 ng ml$^{-1}$ (~2 nM) of EREG and EGF activated ERK to a similar level (Fig S3B and Video 4). Although the cells used in each of these studies are different, these observations imply that the specific activity of EGFR ligands might be markedly different in each study. These potential flaws of bath application of EGFR ligands can be overcome by the re-expression of EGFR ligands in 4KO and QKO cells.

In conclusion, our results demonstrate that there is functional redundancy of EGFR ligands in the propagation of ERK activation waves during collective cell migration of MDCK cells. MDCK cells deficient in all EGFR ligands will provide a platform to examine the physiological function of each EGFR ligand.

# Materials and Methods

## Cells, reagents, antibodies, plasmids, and primers

Reagents, antibodies, plasmids, cell lines and primers are described in the Table S1.

## Cell culture

MDCK and Lenti-X 293T cells were maintained in DMEM (no. 044-29765; Wako) supplemented with 10% FBS (no.172012-500ML; Sigma-Aldrich), 100 unit mL$^{-1}$ penicillin, and 100 mg ml$^{-1}$ streptomycin (no. 26253-84; Nacalai Tesque) in a 5% CO$_2$ humidified incubator at 37°C.

## cDNA cloning of dog EGFR ligands

Total RNA was isolated from MDCK cells using an RNeasy Mini Kit (no. 74104; QIAGEN) according to the manufacturer's instructions. cDNA was reverse transcribed with a PrimeScript II first strand cDNA Synthesis Kit (no. 6210A; Takara Bio). Based on the information at RefSeq (http://www.ncbi.nlm.nih.gov/RefSeq), pairs of PCR primers specific to canine EGF, HBEGF, and EREG were designed with an automated method using Primer3 (https://primer3.org/) as listed in the Table S1. The targeted ORF sequence was amplified by using KOD One PCR Master Mix (no. KMM-101; Toyobo). The cDNA sequences were determined at the DNA Sequencing Facility of the Medical Research Support Center, Kyoto University. The cDNAs of canine TGFα and NRG1 were synthesized by GeneArt (Thermo Fisher Scientific). The cDNA sequences of the cloned growth factors are shown in the Supplemental Data 1.

## Expression plasmids

The cDNAs of EGFR ligands were inserted into pPB-derived vectors (Yusa et al, 2009). ERT2CreERT2 cDNA (Matsuda & Cepko, 2007) was subcloned into pT2A-derived vector (Sumiyama et al, 2010) to generate pT2Aneo-ERT2CreERT2. For rapamycin-inducible activation of Ras, cDNA of Lyn-targeted FRB (LDR) and cDNA of mRFP-FKBP-mSos1-linkercat (Aoki et al, 2011) were subcloned into pPBpuro and pT2Aneo vector, respectively.

## CRISPR/Cas9-mediated KO cell lines

For CRISPR/Cas9-mediated single or multiple knockouts of genes encoding EGFR ligands, EGFR and ADAM17, sgRNAs targeting the exons were designed using CRISPRdirect (Naito et al, 2015), for generating dEGF_2 with a 578 bp deletion, sgRNAs targeting 5′ untranslated region and exon 1 of EGF were designed using CRISPRdirect. Oligo DNAs for the sgRNA were cloned into the lentiCRISPRv2 (Plasmid: no. 52961; Addgene) vector or pX459 (Plasmid: no. 62988; Addgene) vector. The expression plasmids for sgRNA and Cas9 were introduced into MDCK cells by lentiviral infection or electroporation. For lentivirus production, lentiCRISPRv2-derived expression plasmid, psPAX2 (Plasmid: no. 12260; Addgene), and pCMV-VSV-G-RSV-Rev were co-transfected into Lenti-X 293T cells using polyethylenimine (no. 24765-1; Polyscience Inc.). The infected cells were selected with the medium containing the following antibiotics, depending on the drug resistance genes carried by lentiCRISPRv2-derived plasmids; 100 $\mu$g ml$^{-1}$ zeocin (no. 11006-33-0; InvivoGen), 2.0 $\mu$g ml$^{-1}$ puromycin (no. P-8833; Sigma-Aldrich), 200 $\mu$g ml$^{-1}$ hygromycin (no. 31282-04-9; Wako) and/ or 800 $\mu$g ml$^{-1}$ neomycin (no. 16512-52; Nacalai Tesque). For electroporation, pX459-derived expression plasmids were transfected into MDCK cells by an Amaxa Nucleofector II (Lonza). The transfected cells were selected with 2.0 $\mu$g ml$^{-1}$ puromycin. After single cell cloning, genomic DNAs were isolated with SimplePrep reagent (no. 9180; TAKARA bio) according to the manufacturer's instruction. PCR was performed using KOD FX neo (no. KFX-201 TOYOBO) for amplification with designed primers, followed by DNA sequencing.

## Quantitative RT-PCR

Total RNA was extracted using the RNeasy Mini Kit (no. 74104; QIAGEN), and cDNA was reverse transcribed using the High-Capacity cDNA Reverse Transcription Kit (No. 4368814; Thermo Fisher Scientific), according to the manufacturer's instructions. qPCR was performed using the StepOne real-time PCR system (Applied Biosystems) with PowerUp SYBR Green Master Mix (No. A25742; Thermo Fisher Scientific). Based on the information at RefSeq (http://www.ncbi.nlm.nih.gov/RefSeq), pairs of primers specific to canine GAPDH, AREG, BTC, EPGN, NRG1, NRG2, NRG3, and NRG4 were designed with an automated method using Primer3 (https://primer3.org/) as listed in the Table S1. Relative expression levels were calculated using the ΔΔCT method with Canine GAPDH expression as the internal control.

## Expression of FRET biosensors

cDNAs of EKARrEV-NLS were stably expressed either by lentivirus-mediated induction or transposon-mediated gene transfer. The

pPB-derived vectors were co-transfected with pCMV-mPBase (neo-) (Yusa et al, 2009) at a ratio of 4:1 into MDCK cells using the Amaxa nucleofector system (Lonza). Similarly, pT2A-EKAREV-NLS and pCAGGS-T2TP (Kawakami et al, 2004) were co-transfected into MDCK cells by electroporation. The established cell lines are summarized in the Table S1.

### Time-lapse imaging by wide-field fluorescence microscopy

Fluorescence images were acquired essentially as described previously (Aoki & Matsuda, 2009). Briefly, cells cultured on glass-base dishes were observed under an IX83 inverted microscope (Olympus) equipped with a UPlanFL-PH 10×/0.3 (Olympus), a UPlanSApo 20×/0.75 (Olympus), or a UPlanSApo 40×/0.95 objective lens (Olympus), a DOC CAM-HR CCD camera (Molecular Devices), a Spectra-X0 light engine (Lumencor Inc.), an IX3-ZDC laser-based autofocusing system (Olympus), an electric XY stage (Sigma Koki) and a stage top incubator (Tokai Hit). The filters and dichromatic mirrors used for time-lapse imaging were as follows: for FRET imaging, a 438/24 excitation filter incorporated in the Spectra-X light engine, an FF458-Di02-25x36 dichromatic mirror (Semrock), and FF01-483/32-25 and FF01-542/27-25 emission filters (Semrock) for CFP and FRET, respectively.

### Confinement release assay

The confinement release assay was performed as described previously (Hino et al, 2020). To observe collective cell migration of MDCK cells, a Culture-Insert 2 Well (no. 81176; ibidi) was placed on a 35 mm glass-base dish (no. 3911-035; IWAKI) coated with 0.3 mg ml$^{-1}$ type I collagen (Nitta Gelatin). MDCK cells ($3.5 \times 10^4$) were then seeded in the Culture-Insert. 24 h after seeding, the silicone confinement was removed, and the medium was replaced with Medium 199 (11043023; Life Technologies) supplemented with 1% BSA, 100 unit mL$^{-1}$ penicillin, and 100 $\mu$g ml$^{-1}$ streptomycin. Beginning at 30 min after the removal of the silicone confinement, the cells were imaged with an epifluorescence microscope every 2 or 5 min.

### Fluorescence lifetime imaging

For fluorescence lifetime imaging, $3.5 \times 10^4$ MDCK cells were seeded in a Culture-Insert 2 well placed on a 24-well glass-bottom plate coated with 0.3 mg ml$^{-1}$ type I collagen. 24 h after seeding, the silicone confinement was removed, and the medium was exchanged for Medium 199 supplemented with 1% BSA, 100 unit mL$^{-1}$ penicillin, and 100 $\mu$g ml$^{-1}$ streptomycin. 6 h after the removal of the silicone confinement, the cells were imaged to measure the fluorescence lifetime of Turquoise. Lifetime imaging was performed with HC PL APO 20×/0.75 CS2 under a Leica TCS-SP8 microscope (Leica Microsystems GmbH) equipped with a stage top incubator (Tokai Hit), a Lecia HyD SMD detector and a 440 nm ps pulsed diode laser (PDL 800-D; PicoQuant), which pulsed at a frequency of 80 MHz. The band path of emission wavelength was set from 450 to 485 nm. Time-lapse images were acquired every 10 min. The acquisition time for each measurement was 45 s. The amplitude-weighted mean fluorescence lifetimes were calculated in a pixel-by-pixel fashion using fitting

with a multi-exponential ("n-exponential" function) reconvolution with adjustment of the number of components to two according to the manufacturer's protocol (Leica Microsystems GmbH).

### Western blotting with anti–phospho-ERK antibody

For the Western blotting analysis, $3.5 \times 10^4$ cells MDCK cells were seeded in a single well of a Culture-Insert 2 Well that was placed on a glass-bottom dish coated with 0.3 mg ml$^{-1}$ type I collagen. 24 h after seeding, the silicone confinement was removed. The medium was replaced with Medium 199 supplemented with 1% BSA, 100 unit mL$^{-1}$ penicillin, and 100 $\mu$g ml$^{-1}$ streptomycin. 6 h after the removal of the silicone confinement, MDCK cells were lysed with SDS sample buffer containing 62.5 mM Tris–HCl (pH 6.8), 12% glycerol, 2% SDS, 40 ng ml$^{-1}$ bromophenol blue, and 5% 2-mercaptoethanol, followed by sonication with a Bioruptor UCD-200 (Cosmo Bio). After boiling at 95°C for 5 min, the samples were resolved by SDS–PAGE on SuperSep Ace 5–20% precast gels (Wako), and transferred to polyvinylidene difluoride membranes (Merck Millipore) for Western blotting. All antibodies were diluted in Odyssey blocking buffer (LI-COR Biosciences). Proteins were detected by an Odyssey Infrared Imaging System (LI-COR Biosciences).

### Analysis of the cell migration distance in collective cell migration

To measure the migration distance of cells, the FIJI TrackMate plug-in (Tinevez et al, 2017) was applied to the CFP fluorescence images to acquire the track of each cell. The migration distance of each cell was defined by the difference between the abscissae of the first and last time points, 12 or 8 h later. The data analysis was performed by MATLAB.

### Quantification of ERK activity changes in single cells

Time-lapse images of the FRET/CFP ratio were generated after background subtraction by using Metamorph software (Molecular Devices) as described previously (Aoki & Matsuda, 2009). For single cell analysis of the ERK activity change, the images were analyzed by using the FIJI plug-in (Schindelin et al, 2012). On CFP images, the following commands were applied sequentially: "8-bit," "Subtract Background…" with the rolling size of 50 pixels, "Make Binary" with the Otsu method, "Watershed," "Erode," and "TrackMate" for tracking of each cell. The time-series data of the coordinates of each cell and the FRET/CFP ratio were processed with a Savitzky–Golay filter to reduce the noise.

The coefficient of duration $\omega_i$ and half of amplitude $A_i$ were fitted as follows:

$$f_i(t) = \begin{cases} A_i \left( 1 + \sin\left( \frac{\pi}{2} + \frac{t - \theta_i}{\omega_i} \right) \right) & if -\pi < \frac{t - \theta_i}{\omega_i} < \pi \\ 0 & otherwise \end{cases}, \qquad (1)$$

where $i$ is the pulse index, t is the timepoint, and $\theta_i$ is the timepoint of the pulse peak.

The programs used for sine curve fitting of MATLAB are shown in the Supplemental Data 1.

### Analysis of ERK activation waves with heat maps

To determine the sample directions of ERK activation waves, heat maps of ERK activity were obtained by interpolating the signals in regions between the nuclei of MDCK cells in the FRET/CFP ratio images (Hino et al, 2020). The heat maps of ERK activity were analyzed by PIV using a free MATLAB-toolbox, MatPIV (Sveen, 2004), with a 128 $\mu$m window size and a 75% window overlap. The directions of the calculated velocity vectors were obtained as the sample directions. To obtain the kymographs of FRET/CFP ratios, these values were averaged along the y-axis in a defined region of the images, providing an intensity line along the x-axis. The operation was repeated for the respective time points, and the intensity lines were stacked along the y-axis for all time points. The ERK activation waves were detected and counted after binarizing by using the Regionprops (Image Processing Toolbox) function in MATLAB.

### Induction of ERK activation waves with rapamycin-inducible mSos1 translocation

The rapamycin-inducible mSos1 translocation and Ras activation were reported previously (Aoki et al, 2011). MDCK-WT-EKARrEV-NLS-LDR-mRFP-FKBP-mSos1-linkercat cells were seeded in a well of the Culture-Insert 1 well placed on a 24-well glass-bottom plate (no. 5826-024, IWAKI). MDCK-WT-EKARrEV-NLS cells or MDCK-4KO-EKARrEV-NLS cells were seeded in the outside of the silicone confinement. After 24 hours incubation, the silicone confinement was removed. Further incubation for 48 h allowed the cells to fill the gap between the two cell populations. During observation of the interface between the two cell populations, rapamycin was added to a final concentration of 250 nM. To examine the propagation of ERK activation waves, heat maps of ERK activity were obtained by interpolating the signals in regions between the nuclei of cells in the FRET/CFP ratio images.

### Analysis of the area of ERK activation waves

To examine the area of the ERK activation waves, heat maps of ERK activity were obtained by interpolating the signals in regions between the nuclei of cells in the FRET/CFP ratio images. To obtain a binarized image of the interpolated FRET/CFP ratio images, first these ratio images were denoised, and then a locally adaptive threshold was computed for a 2D grayscale image by using "adaptthresh" in MATLAB, after which the following arguments were applied sequentially: "sensitivity" of 0.65 and "ForegroundPolarity" with "dark." After obtaining binarized images, wave areas of each frame in each cell line from six to 9 h after releasing the silicone confinement was counted.

### Analysis of the duration of the transient ERK activation after addition of recombinant human EGFR ligands

To examine the duration of the transient ERK activation after application of recombinant human EGFR ligands, time of ERK activation recovering to the half-maximal were calculated. The definition of half-maximal is the average of ratio value before EGFR ligands adding (basal) and just after adding (maximum) in each experiment.

### Measurement of growth rate

For the growth rate measurement, $5 \times 10^4$ MDCK-5KO-EKARrEV-NLS-loxP-NRG1 or MDCK-5KO-EKARrEV-NLS-loxP-NRG1-CreERT2 or $2.5 \times 10^4$ MDCK-5KO-EKARrEV-NLS-loxP-NRG1-CreERT2 and $2.5 \times 10^4$ MDCK-WT cells were seeded in a six-well plate (No. 140675; Thermo Fisher Scientific) and cultured in DMEM containing 10% FBS. After 24 h incubation, 1 $\mu$M 4-hydroxytamoxifen (no. 579002; Sigma-Aldrich) or DMSO (no. 13445-74; Nacalai Tesque) or 1 $\mu$M 4-hydroxytamoxifen and 10 ng/ml EGF (no. E9644; Sigma-Aldrich) was added. Fluorescence images of fixed locations were acquired with a UPlanFL-PH 10x/0.3 objective lens (Olympus) every 24 h. Images were binarized to count the number of nuclei by FIJI.

### Characterization of EKARrEV-NLS

We first established MDCK cells stably expressing EKARrEV-NLS or the prototype EKAREV-NLS. Then, $2 \times 10^4$ cells were seeded in a well of a 24-well glass-bottom plate coated with 0.3 mg ml$^{-1}$ type I collagen. After 24 h, the medium was replaced with Medium 199 supplemented with 1% BSA, 100 unit mL$^{-1}$ penicillin, and 100 $\mu$g ml$^{-1}$ streptomycin. Fluorescence images were acquired with an UPLSAPO 20X objective (Olympus) every 2 min. During imaging, EGF (no. E9644; Sigma-Aldrich) or trametinib (no. T-8123; LC Laboratories) was added to a concentration of 100 ng ml$^{-1}$ or 200 nM final, respectively. After applying autothreshold, the fluorescence intensities of each nucleus were quantified to obtain FRET/CFP values as described previously (Aoki & Matsuda, 2009).

### Statistical analysis

Probability ($P$) values were determined by using the T.TEST function of Microsoft Excel with two-tailed distribution and two-sample unequal variance.

## Data Availability

The raw data from this publication have been deposited to the Systems Science of Biological Dynamics repository database. https://ssbd.riken.jp/repository/174/.

## Supplementary Information

## Acknowledgements

We thank the members of the Matsuda Laboratory for their helpful input and encouragement, K Hirano and K Takakura for their technical assistance, Takahiko Matsuda for pCAG-ERT2CreERT2, and the Medical Research Support Center of Kyoto University for in vivo imaging. This work was supported by the Kyoto University Live Imaging Center. Financial support was provided in the form of JSPS KAKENHI grants (nos. 18K07066 to K Terai, 20H05898 to M Matsuda, and 19H00993 to M Matsuda), a JST CREST grant (no. JPMJCR1654), a

Moonshot R&D grant (no. JPMJPS2022-11 to M Matsuda), and funds from the Fugaku Foundation (to M Matsuda).

## Author Contributions

S Lin: conceptualization, resources, data curation, software, formal analysis, validation, investigation, visualization, methodology, and writing—original draft, review, and editing.

D Hirayama: resources, data curation, and formal analysis.

G Maryu: data curation, visualization, and methodology.

K Matsuda: resources and data curation.

N Hino: resources, data curation, software, formal analysis, and methodology.

E Deguchi: resources, data curation, and formal analysis.

K Aoki: conceptualization, investigation, and methodology.

R Iwamoto: resources.

K Terai: conceptualization, resources, formal analysis, investigation, and methodology.

M Matsuda: conceptualization, resources, data curation, software, formal analysis, supervision, funding acquisition, validation, investigation, visualization, methodology, project administration, and writing—original draft, review, and editing.

## Conflict of Interest Statement

The authors declare that they have no conflict of interest.

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
