## [Reviewer comments · Life Science Alliance]

Life Science Alliance

Redundant roles of EGFR ligands in the ERK activation waves during collective cell migration

Shuhao Lin, Daiki Hirayama, Gembu Maryu, Kimiya Matsuda, Naoya Hino, Eriko Deguchi, Kazuhiro Aoki, Ryo Iwamoto, Kenta Terai, and Michiyuki Matsuda

DOI: <https://doi.org/10.26508/lsa.202101206>

Corresponding author(s): *Michiyuki Matsuda, Kyoto University*

Review Timeline:

Submission Date:	2021-08-22
Editorial Decision:	2021-08-23
Revision Received:	2021-09-27
Editorial Decision:	2021-10-07
Revision Received:	2021-10-07
Accepted:	2021-10-08

Scientific Editor: *Eric Sawey, PhD*

Transaction Report:

Please note that the manuscript was reviewed at *Review Commons* and these reports were taken into account in the decision-making process at *Life Science Alliance*.

Review
COMMONS

August 23, 2021

Re: Life Science Alliance manuscript #LSA-2021-01206-T

Dr. Michiyuki Matsuda
Graduate School of Biostudies, Kyoto University
Laboratory of Bioimaging and Cell Signaling
Yoshida-Konoe-Cho, Sakyo-ku
Kyoto, Kyoto-fu 606-8501
Japan

Dear Dr. Matsuda,

Thank you for submitting your manuscript entitled "Redundant roles of EGFR ligands in the ERK activation waves during collective cell migration" to Life Science Alliance. We invite you to re-submit the manuscript, revised according to your Response to the Reviewers.

Thank you for this interesting contribution to Life Science Alliance. We are looking forward to receiving your revised manuscript.

Sincerely,

Eric Sawey, PhD
Executive Editor
Life Science Alliance
<http://www.lsa-journal.org>

- A letter addressing the reviewers' comments point by point.
- An editable version of the final text (.DOC or .DOCX) is needed for copyediting (no PDFs).
- High-resolution figure, supplementary figure and video files uploaded as individual files: See our detailed guidelines for preparing your production-ready images, <https://www.life-science-alliance.org/authors>
- Summary blurb (enter in submission system): A short text summarizing in a single sentence the study (max. 200 characters including spaces). This text is used in conjunction with the titles of papers, hence should be informative and complementary to the title and running title. It should describe the context and significance of the findings for a general readership; it should be written in the present tense and refer to the work in the third person. Author names should not be mentioned.
- By submitting a revision, you attest that you are aware of our payment policies found here: <https://www.life-science-alliance.org/copyright-license-fee>

B. MANUSCRIPT ORGANIZATION AND FORMATTING:

Manuscript number: RC-2021-00857

Corresponding author(s): Michiyuki, Matsuda

1. General Statements [optional]

First of all, we would like to thank the editor and all reviewers for the effort to evaluate our manuscript in this challenging era of COVID-19.

Reviewer #1

(Significance): Overall, this manuscript is very clear and easy to follow. The manuscript could be improved by making the following changes:

We thank the reviewer for the favorable comment and have revised the manuscript according to the suggestions.

Line 47 in Abstract should read "Aiming for" not "Aiming at".

We have corrected the typo as suggested (**Line 57**).

Some in the field call fluorescence lifetime microscopy "FLIM", you could adopt the same wording in your manuscript to attract more readers.

We have included "FLIM" according to the reviewer's suggestion (**Line 146**).

The names QKO and 4KO are a bit confusing. Could the authors please change the naming of the knockout cells so that readers understand that QKO and 4KO are two separate cell types? Perhaps instead of 4KO use FKO for "full knockout" or something similar. The 5KO line might also need to be named something else if you change to FKO.

We have discussed this issue with the co-authors, but could not reach a better idea. Instead of changing the names, we have modified the explanation for the 4KO cell line as follows:

*"After finding marked suppression of the ERK activation waves in QKO, all four genes were knocked out simultaneously by introducing four gRNAs. The new clone deficient from the four EGFR ligand genes was designated as 4KO." (**Line 134**)*

Figure 1D, the images should be presented using the same scale for both the EKAREV and EKARrEV constructs so that they can be directly compared.

Because the basal FRET/CFP ratio is significantly different between EKAREV-NLS and EKARrEV-NLS, the changes of FRET/CFP ratio during mitosis become unclear if we applied the same scale. This figure is prepared to show the difference of the reactivity to Cdk1 during mitosis; therefore, we believe the current scale is better for presentation.

For Fig 1F, 3 individual experiments should be conducted to confirm results.

We have repeated this experiment and modified **Fig. 1F** and the legend accordingly (**Line 755**).

For Fig 1G, could the authors please show the original western blot data in full rather than just the densitometry graphs?

We have included one of the three sets of the blot image used to prepare Fig. 1G as supplementary **Fig. S1C (Line 899)**.

The authors should explain the origin/phenotype of MDCK cells for those who are not familiar with the cell line.

According to the suggestion, in the first paragraph of the introduction, we have introduced MDCK cells and deleted the sentences related to the ERK activation waves during development for brevity (**Line 67-**).

“Madin-Darby Canine Kidney (MDCK) cells, which hold clear apical-basolateral polarity and clear cell junctions (Dukes et al., 2011), are frequently used as a model of collective cell migration (Reffay et al., 2014; Das et al., 2015). The EGFR (epidermal growth factor receptor)-ERK (extracellular signal regulated kinase) signaling cascade plays a pivotal role in the collective cell migration of various cell types including MDCK cells (Friedl and Gilmour, 2009; Yarden and Sliwkowski, 2001).”

The authors should give a future outlook/direction for future experimentation to further confirm redundancy in EGF ligands in the propagation of ERK activation waves.

We have added the following paragraph in the discussion section (**Line 288 -**).

“Currently, we have not succeeded in eliminating the ERK activation waves by the knockout of EGFR ligand genes. This failure is because we could not examine the effect of Nrg-1 in 4KO cells due to the growth arrest (**Fig. 5B**). To further pursue this issue, we probably need to establish MDCK cell variants that grow without input from receptor-type tyrosine kinases. Augmentation of cell survival signals such as the PI-3 kinase pathway may help to establish such cell lines.”

Reviewer #2

(Evidence, reproducibility and clarity):

Lin et al. address the mechanisms underlying ERK signaling waves in epithelial cells. While it is known that ADAM17 is critical to process EGFR ligands, the specific or redundant roles of different ligands remains an open question. First the authors generate a modified ERK FRET sensor with reduced cross-reactivity to CDK1 in MDCK cells and systematically knockout EGF, HBEGF, TGF α and EREG (the highest expressed ligands in MDCK cells). The authors use live cell imaging of ERK activity upon release from confinement and find that all ligands contribute to ERK signaling waves. While differences in basal signaling and other dynamic features are found in individual knockouts, only the quadruple KO cells show a significant decrease in ERK waves. To determine if the 4KO cells are defective in wave propagation (as opposed to wave initiation), the authors coculture 4KO cells with an inducible cell line and conclude that 4KO cells are unable to propagate waves. Individual EGFR ligands are then restored in 4KO cells, and EGF, TGF α , and EREG, but not HBEGF, can rescue ERK activity waves. Finally, the authors attempt to eliminate all ERK activation waves by deletion of Nrg1 but find that it is essential in 4KO cells. The paper is well-written and technically sound. The use of genetics is particularly impressive but the lack of major discoveries dampens the enthusiasm. Additional efforts to mechanistically define wave initiation and wave propagation would significantly improve the impact of the manuscript. Moreover, some of the conclusions are not fully supported by the data and require further experimentation and/or analysis.

We admit that significant redundancy of function among the EGFR ligands and their essential roles in cell growth prevent us from obtaining clear results. Considering the importance of EGFR ligands in biology, we believe, our observation gives invaluable suggestions to those who wish to clarify the roles played by EGFR-family protein in other biological contexts.

1. There are conflicts with some of the conclusions made about ligands. dEGFR cells have basal ERK activity as high as WT which argues against EGF being responsible for basal ERK activity. Further, basal ERK activity was not rescued by restoration of EGF in the 4KO-EGF cells. The authors should address this discrepancy.

We agree that some new questions have arisen from our observations. One of them is the decreased basal ERK activity in dEGF cells and the failure to restore the basal ERK activity by the EGF re-expression. We had used two independent dEGF clones obtained by different experiments to confirm the results. Furthermore, during this revision, we established dEGF cell lines, in which most of the exon 1 of the *Egf* gene is deleted. Again, we observed that *Egf* deficiency decreases basal ERK activity (**Fig. S1D**). Therefore, we are confident about our findings and added a paragraph to discuss this enigma (**Lines 320-330**). The result section was also modified to refer to this new dEGF clones (**Lines 149-151**).

2. Besides the ones genetically disrupted in this work, other EGFR ligands seem to play functional roles given that dEGFR cells less migration and fewer ERK waves than 4KO cells. The authors could test if other ligands are upregulated in 4KO cells to compensate. On a similar note determining whether ADAM17 deficient cells are more similar to 4KO cells or dEGFR cells could provide some insight.

According to the reviewer's suggestion, we conducted qPCR analysis of EGFR ligands and neuregulins in mutant cell lines (**Fig. S1B**). We did not find marked upregulation of their expression levels. We also developed an *Adam17* deficient cell line, dADAM17. The data are

included in **Fig. 1E-G, Fig. 2, Fig. 3B, and Fig. 3C**. The result, discussion, and figure legend sections are modified accordingly. Related to the first question, the phenotype of *Adam17* deficiency cannot be understood straightforwardly. *Adam17* deficiency suppressed ERK activation waves and decreased the average and basal ERK activity. The decrease of average and basal activity is the phenotype of EGF, which is not the substrate of Adam17. The effect of *Adam17* deficiency is also discussed in the discussion section (**Lines 320-330**).

3. *The interpretation of the RA-SOS coculture experiments is confusing. Based on the author's reasoning, I would expect ADAM17 shedding in the RA-SOS cells to trigger signaling at the interface of both WT and 4KO cells but the 4KO should be unable to propagate the wave farther away from the interface. This does not seem to be the case. Do RA-SOS ADAM17KO cells still trigger waves of ERK signaling in the WT cells? Do ADAM17KO cells behave as the 4KO cells in this coculture system?*

We previously performed similar experiments using *Adam17*-deficient MDCK cells expressing 2paRAF, a photo-activatable RAF protein. We found that light-induced ERK activation did not efficiently trigger cell contraction of the *Adam17*-deficient MDCK cells and thereby ERK activity propagation in surrounding neighbors (Panels A and B). This effect of *Adam17* deficiency on cell contraction is independent of EGFR activity because we found that an EGFR inhibitor PD153035 did not suppress cell contraction triggered by ERK activation but ADAM inhibitor marimastat treatment did (Figures C and D). These results indicate that *Adam17*-deficient MDCK cells cannot propagate ERK activation even to the wild-type cells. There are many substrates of ADAM17. Because the interpretation of this data is not straightforward, and because ADAM17 is not the main issue of this paper, we would like to refrain from including this data into the present paper.

For the reviewer. (A) The boundary assay using WT or ADAM17 KO cells. Displacement of the boundary, indicative of cell contraction, is plotted over time after blue light exposure. The lines represent the average with SDs. $n = 9$ from two independent experiments. (B) The dot plots represent the boundary displacement at 120 min after blue light exposure. The bars indicate the average with SDs. (C) The boundary assay in the presence of 0.1% DMSO, 1 µM PD153035 (EGFR inhibitor), or 10 µM Marimastat (ADAM/MMP inhibitor). The lines represent the average with SDs. $n = 17$ (DMSO) and 18 (PD153035, Marimastat) from three independent experiments. The Data of DMSO and PD153035 was adapted from Hino et al., Dev Cell, 2020 (Fig. S2B). (D) The dot plots represent the boundary displacement at 120 min after blue light exposure. The bars indicate the average with SDs.

4. The authors propose that Nrg1 is responsible for ERK waves in QKO, 4KO, dEGFR, and 4KO-EGF cells but are limited in testing this due to Nrg1 being essential in 4KO cells. First, Nrg1 should have been deleted in TKO cells to confirm that it is only essential in the absence of the four EGFR ligands. Additionally, Nrg1 could be knocked out in 4KO-EGF cells to demonstrate the claim that EGF-induced ADAM17 cleavage of Nrg1 is responsible for ERK waves.

We would like to argue against the reviewer's idea. We do not think the deletion of Nrg1 in the TKO cells abolishes the ERK activation waves because EREG in TKO cells could transmit the waves. To overcome the problem of cell growth, we attempted to provide EGFR ligands in trans. We found that 5KO-loxP-NRG1 cells can grow in the presence of 4-OHT, when they are co-cultured with wild type MDCK cells (Fig. 5C). However, the exogenous addition of EGF did not work (Fig. 5D). The result section has been modified to refer to this observation (**Lines 268-275**).

5. The authors state that ERK activation waves are important for collective migration and seek to understand the roles of each EGFR ligand, but despite measuring migration and properties of ERK activity, there is very little analysis or commentary on the relationship between the two. The ability of HB-EGF to restore migration without ERK waves suggests that waves are not required per se. It is interesting to note that with restoration of ligands, migration is higher than WT but ERK activity is lower.

We refrained from spending much space about the essential role of ERK activation waves in collective cell migration because several papers have already described this issue. We should have spent more space emphasizing that the collective cell migration comprises at least two different phenomena—the migration of leader cells and that of the follower cells (**Fig. 2B**). The ERK activation waves are essential for the migration of follower cells but not the leader cells. In 4KO cells, migration of both the leader cell and the follower cell are impaired. We have shown that expression of HBEGF restores the leader cell migration, but have not examined the migration of the follower cells. We have explicitly stated that we measured the migration of the leader cells.

“Since the migration of the leader and submarginal cells was accelerated by the HBEGF expression to the level of other EGFR ligands (Fig. 4B), functional HBEGF was expressed in 4KO-HBEGF cells.” (**Line 306**)

6. It is suggested that the total amount of EGFR ligands may be the primary determinant of migration, but deletion of TGF α alone causes a significant decrease in migration comparable to the DKO cells. TGF α has the lowest expression of the four ligands studied but is the only ligand to have a significant impact on migration in the single knockout context, which disagrees with that conclusion.

The p value of dTGF α was 0.017. We hesitate to discuss the significance of this finding. We are currently characterizing each EGFR ligand and have already found unique properties of each EGFR ligand. For example, TGF α is efficiently cleaved by the basal ADAM17 activity, which probably renders TGF α the primary EGFR ligand to promote the migration of the leader and submarginal cells. But, in this paper, we would like to refrain from spending spaces on the migration of the leader and submarginal cells.

Other:

1. In Fig. 1G, the normalization of all WT pERK samples to 1 artificially lowers the variance to zero when performing the T-test.

To compare immunoblotting data derived from independent experiments, the signals must be normalized to the control. We believe the use of pERK/ERK of the wild type cells as the control is reasonable for this experiment.

2. Fig. S3B needs clarification that the WT (black) and 4KO (green) did not receive a stimulus.

Following statement has been included as suggested.

“WT (black) and 4KO (green) are controls without any EGFR ligands.” **(Line 922)**.

(Significance): While it is known that ADAM17 is critical to process EGFR ligands, the specific or redundant roles of different ligands remains an open question. The authors find that all ADAM17 ligands contribute to ERK signaling waves but may have specific contributions to other phenotypes. This work would be of interest to the signaling dynamics, epithelial and developmental biology communities.

We thank the reviewer for the favorable comment.

Reviewer #3

(Evidence, reproducibility and clarity):

This manuscript seeks to clarify the mechanisms that underlie traveling "waves" of ERK activity that occur in monolayers of migrating epithelial cells. A combination of live cell imaging with ERK activity biosensors and CRISPR-mediated knockouts for autocrine regulators are used to dissect the factors that make these waves possible. The authors utilize the MDCK cell line, which shows very prominent wave behavior, and they perform an impressive number of knockouts to eliminate the most abundant autocrine EGFR ligands. They also introduce a novel ERK FRET reporter, which is less sensitive to off-target phosphorylation by Cdk1. Analysis of ERK biosensor data from the knockouts shows that knockout of all four main EGFR ligands is needed to substantially reduce the amplitude of ERK waves, although it does not completely eliminate it. Re-expression of any of the four ligands, with the exception of HBEGF, restores strong ERK waves. Application of the same ligands in solution restores migration but not the ERK waves.

Overall, this study is carried out with a high degree of rigor and technical excellence, with clear reporting of experimental details and replication. The writing and figures are very clear, and there are no obvious technical problems. However, there are some areas in which the strength and clarity of the conclusions could be strengthened by relatively simple experiments.

We thank the reviewer for the favorable comment. We have added some data to reinforce our proposal.

****Major:****

The experiments in Fig. 5 are undertaken with the purpose of assessing whether NRG acts as an additional ligand that mediates the residual ERK waves in 4KO/QKO cells. However, this question is never addressed in the NRG/4KO cells. While it might be challenging due to the proliferative defect, it seems important to attempt this experiment in some way; measuring the ERK waves for these cells would establish whether all of the critical autocrine factors have been identified. Can the proliferation be rescued by application of high amounts of growth factors?

To overcome the problem of the cell growth, we attempted to provide EGFR ligands in trans. We found that *Nrg1* can be deleted from 4KO cells, when they are co-cultured with wild type MDCK cells. However, exogenous bath application of EGFR ligands did not work. The data have been included in **Fig. 5C and D**. The result section has been modified accordingly (**Lines 268 – 275**).

The bath exposure to EGFR ligands shown in Fig. S3A is an important experiment, but it is surprising that ERK signaling is not maintained under these conditions. Is this due to depletion of the added ligands, perhaps locally? Or is the intermittent nature of paracrine signaling needed to maintain ERK activity? These possibilities could be distinguished by checking whether the added EGF or the other ligands are depleted after several hours, or by restimulating with a new bolus of ligand after several hours.

According to the reviewer's suggestion, we restimulated cells with EGFR ligands (**Fig. S3E**). We observed modest ERK activation by EGFR ligands, suggesting partial recovery of the plasma membrane expression of EGFR. We have included the statement on this finding (**Line 245**).

Minor (I think this is an important point overall, but it is outside of the scope of the paper as defined by the authors, which is focused on the ERK waves rather than how the waves relate to migration):

1. The connection between ERK activity and migration is somewhat confusing. It would be helpful to show the dose sensitivity of migration to a MEK or ERK inhibitor. Are other pathways downstream of EGFR such as PI3K involved in the autocrine-mediated migration? This could also be established with the appropriate inhibitors.

We should have spent more space to emphasize that the collective cell migration is comprised of at least two different phenomena. The migration of leader/submarginal cells and the follower cells. The migration of the leader/submarginal cells depend on ERK activity to some extent, but not entirely, probably because of the contribution of other pathways such as the PI3K pathway. However, the focus of our work is the ERK activation waves; therefore, we would like to refrain from adding more data on the migration of the leader/marginal cells. To emphasize that we are more interested in the propagation of ERK activation waves, we add a few sentences in the result section (**Line 177 - 180**).

“We previously reported that ERK activation waves from the leader cells is indispensable for the directed migration of the follower cells (Hino et al., 2020; Aoki et al., 2017). Therefore, the effect of EGFR ligand deficiency on the propagation of the ERK activation waves was examined hereafter.”

(Significance): This study definitively establishes the role of 4 EGFR ligands in the generation of ERK activity waves in MDCK cells. While other studies, including some from the senior author's lab, have strongly indicated that EGFR autocrine signaling is important for these waves, this study goes further in comparing the roles of these ligands using knockouts to unambiguously establish the autocrine factors involved. Others who use this common experimental system (MDCK) to study epithelial dynamics will find this study of great interest. A wider audience of those who work on EGFR-mediated signaling will also find the data quite fascinating as an example of the complex relationship between ERK activation and its downstream effects. The technical excellence of the paper will make it a must-read for those in these fields. However, there are some factors that limit the scope of the significance. MDCK cells are an important experimental model system but differ in substantial ways from other epithelial cells, particularly in the expression of EGFR ligands. Because different ligands such as amphiregulin dominate in other systems (as noted by the authors, and PMID 27405981), the ability to extrapolate from these findings to other cell types is somewhat limited. Also, the paper avoids addressing the major question of how ERK waves relate to collective migration rate. From the data presented it is clear that this relationship is complex; for example, bath application of the ligands restores a high migration rate but not ERK waves. Given this lack of a clear relationship it is an understandable decision to leave this question for future work; however this does limit the conclusions that can be drawn from the study.

We completely agree with the reviewer's view. It is uncertain to what extent the observation with MDCK cells can be generalized to other cell types. We also admit that the conclusion is not very simple because EGFR signaling is required for various cellular functions including cell survival and migration. Even though the gene editing becomes so easy, it is still labor consuming work to knock out many genes in a single cell line with extensive characterization.

We believe the data shown in our work will provide a basis for the understanding of EGFR ligands.

October 7, 2021

RE: Life Science Alliance Manuscript #LSA-2021-01206-TR

Dr. Michiyuki Matsuda
Kyoto University
Department of Pathology and Biology of Diseases
Graduate School of Medicine
Kyoto, Kyoto-fu 606-8501
Japan

Dear Dr. Matsuda,

Thank you for submitting your revised manuscript entitled "Redundant roles of EGFR ligands in the ERK activation waves during collective cell migration". We would be happy to publish your paper in Life Science Alliance pending final revisions necessary to meet our formatting guidelines.

- please add the Twitter handle of your host institute/organization as well as your own or/and one of the authors in our system
- please use the [10 author names, et al.] format in your references (i.e. limit the author names to the first 10)
- please add your main, supplementary figure, video, and table legends to the main manuscript text after the references section

A. FINAL FILES:

B. MANUSCRIPT ORGANIZATION AND FORMATTING:

Sincerely,

Reviewer #1 (Comments to the Authors (Required)):

The authors have addressed all my comments. I appreciate all the efforts to include dADAM17 data.

October 8, 2021

RE: Life Science Alliance Manuscript #LSA-2021-01206-TRR

Dr. Michiyuki Matsuda
Kyoto University
Department of Pathology and Biology of Diseases
Graduate School of Medicine
Kyoto, Kyoto-fu 606-8501
Japan

Dear Dr. Matsuda,

Thank you for submitting your Research Article entitled "Redundant roles of EGFR ligands in the ERK activation waves during collective cell migration". It is a pleasure to let you know that your manuscript is now accepted for publication in Life Science Alliance. Congratulations on this interesting work.

DISTRIBUTION OF MATERIALS:

Again, congratulations on a very nice paper. I hope you found the review process to be constructive and are pleased with how the manuscript was handled editorially. We look forward to future exciting submissions from your lab.

Sincerely,
